# Etiological characteristics of 3,691 cases of microbial keratitis: an 8-year longitudinal study

Yi Xu,[1] Bianjin Sun,[2] Yangyang Shen,[1] Huijing Xu,[1] Yunfeng Gu,[1] Liping Mao,[1] Ying Liang,[1] Qingsong Lu,[1] Meiqin Zheng[1]

**ABSTRACT** The etiological spectrum of microbial keratitis exhibits significant regional variability. However, existing domestic and international resistance monitoring systems do not adequately address the specific needs of clinical practice. We analyzed a cohort of 3,691 patients diagnosed with microbial keratitis in the southern region of Zhejiang Province, China between 2016 and 2023. The patients' ages ranged from 2 to 93 years (979 men and 403 women). The microbial culture positivity rate is relatively low (38.72%). Trauma and foreign body entry emerged as significant risk factors. Mixed infections accounting for 4.85% posed challenges for diagnosis and treatment. Filamentous fungi, predominantly *Fusarium* spp., dominated the microbial landscape. Prominent bacterial pathogens included *Pseudomonas aeruginosa*, *Streptococcus pneumoniae*, *Staphylococcus epidermidis*, and *Staphylococcus aureus*. *Acanthamoeba spinosa* was an important pathogen affecting the cornea in this region. The observed resistance patterns emphasized the urgency for alternative therapeutic strategies targeting resistant gram-positive (e.g., erythromycin, penicillin, oxacillin) and -negative (e.g., trimethoprim–sulfamethoxazole, tetracycline) bacteria as well as refractory fungi, such as *Candida albicans* (voriconazole-resistant) and *Aspergillus flavus* (resistant to itraconazole and amphotericin B). Resistance to ceftazidime, meropenem, and erythromycin exhibited a slight upward trend, diverging from the overall bacterial resistance trend observed in China. High isolation rates of methicillin-resistant *S. aureus* and macrolide-resistant *S. pneumoniae* underscored the need for enhanced infection control measures and targeted interventions against these resistant pathogens. This study elucidated the evolving patterns of antibiotic resistance among ocular isolates in the region, providing a critical foundation for the effective application of antimicrobial therapies in clinical practice.

**IMPORTANCE** Currently, our region does not possess extensive monitoring data regarding antibiotic resistance trends in ocular isolates, especially those derived from corneal infections. This study addresses a critical component of ophthalmic microbiology by analyzing long-term data to identify trends in etiological agents and their clinical implications. It aimed to fill the void in epidemiological data on ocular isolates within our region, providing scientific insights essential for the comprehensive monitoring of ocular microbial drug resistance.

**KEYWORDS** keratitis, eye trauma, filamentous fungi, multidrug resistance

The terms "microbial keratitis," "infective keratitis," and "suppurative keratitis" refer to suppurative infections of the cornea (1). Microbial keratitis (MK) is an ocular emergency primarily caused by bacteria, fungi, viruses, or protozoa and characterized by a high incidence rate and a poor visual prognosis. This condition is prevalent worldwide, particularly in South, Southeast, and East Asia, with an estimated annual incidence exceeding 2 million cases. MK is the fifth leading cause of blindness globally, accounting

**Peer Reviewers** Innocent Afeke, University of Health and Allied Sciences, Ho, Ghana; Suleiman Fawaz al Obeid, Security Forces Hospital, Riyadh, Saudi Arabia

Address correspondence to Meiqin Zheng, zmq@eye.ac.cn.

The authors declare no conflict of interest.

See the funding table on p. 15.

for more than 5% of all blindness cases (2). The latency period of MK ranges from 2 to 30 days, with the disease progressing rapidly after onset. Clinical manifestations include corneal ulcers, epithelial defects, and interstitial infiltration. Without prompt and effective treatment, MK can progress swiftly, leading to corneal perforation, eye atrophy, blindness, and other adverse outcomes that severely affect the daily lives and work of patients. Therefore, aggressive treatment and prompt control of MK are crucial for improving patient prognosis (3).

The human ocular system possesses a complex structure, with notable disparities in the spectra of pathogens isolated from various physiological components (4). The etiological spectrum of MK exhibits significant regional variability (5). However, existing domestic and international resistance monitoring systems primarily provide broad analyses of ocular pathogens, which do not adequately address the specific needs of clinical practice. Currently, our region lacks comprehensive, long-term, large-scale data on the infection patterns and drug resistance surveillance of corneal isolates. Therefore, this study was conducted to analyze the distribution and resistance patterns of pathogens causing MK in Southern Zhejiang over the past 8 years. The objectives were to identify resistance trends, evaluate treatment efficacy, and provide meaningful guidance for clinical strategies.

## MATERIALS AND METHODS

### Sample source

A retrospective collection of the samples of patients with MK was conducted at the Eye Hospital of Wenzhou Medical University between 1 January 2016 and 31 December 2023. We included patients with confirmed positive cultures for bacteria, fungi, and parasites, adhering to the rigorous standards outlined in the "Expert Consensus on the Clinical Diagnosis and Treatment of Infectious Keratopathy" (6). Our inclusion criteria rigorously selected patients presenting with moderate-sized corneal ulcers (exceeding a diameter of 1 mm) or atypical corneal ulcers, all of whom underwent comprehensive microbiological assessments, including corneal scraping microscopy (Gram staining), microbial culture, and thorough susceptibility testing. In accordance with the "Expert Consensus on Bacteriological Examination of Infectious Eye Diseases" (7), the sampling procedures were executed by trained personnel, who used a No. 15 blade to delicately scrape the base and edge tissue of the ulcer for sampling. These samples were promptly inoculated onto the appropriate media at the bedside, ensuring optimal handling and minimizing the risk of contamination. To preserve the integrity of the findings, rigorous exclusion criteria were applied. Duplicate samples from the same patient and strains suspected of contamination were excluded. Contaminated strains were identified by colonies that failed to grow along the inoculation streak or isolated strains that did not align with clinical infection symptoms or auxiliary test results, such as confocal microscopy and smear examination. These measures were implemented to ensure the accuracy and reliability of the analysis. The study protocol was approved by the Research Ethics Committee of Wenzhou Medical University (Approval Number: 2024-155-K-133), reaffirming our commitment to ethical conduct and the protection of patient privacy. Informed consent was not required, as we utilized anonymous, preexisting data from patient records, which did not involve direct interaction with individuals or collection of new data.

### Pathogen culture and identification

Clinical specimens were divided into two groups for inoculation, with one group placed on chocolate agar (35℃, 5% $CO_2$) and the other on potato dextrose agar (28℃). In cases of suspected *Acanthamoeba* keratitis, non-nutrient agar with *Escherichia coli* was used. Cultures were incubated rigorously for 1 week.

Positive bacterial cultures were identified using Autof MS600 matrix-assisted laser desorption/ionization time-of-flight mass spectrometry and the BioMérieux ATB System,

with genus-level identification when necessary, adhering to International Organization for Standardization (ISO) 15189 standards. Fungal identifications were primarily conducted at the genus level, utilizing colony morphology, lactophenol cotton blue staining, and mass spectrometry. Unidentified fungi were classified as filamentous or yeast-like based on microscopy.

## Antimicrobial susceptibility testing

Antimicrobial susceptibility testing was ISO 15189-accredited, adhering to the Clinical and Laboratory Standards Institute (CLSI) guidelines. Minimum inhibitory concentrations (MICs) for bacteria were determined using a microbroth dilution method, whereas MICs for fungi were assessed using both E-test and microbroth dilution methods. The selection of antimicrobial agents was based on the CLSI M100 and M45 guidelines supplemented by standard ophthalmic medications. Breakpoints were established based on the 2023 CLSI guidelines to ensure clinical relevance.

## Quality control strains

American Type Culture Collection (ATCC) reference strains, including *Staphylococcus aureus* (ATCC29213), *E. coli* (ATCC25922), *Pseudomonas aeruginosa* (ATCC27853), *Candida parapsilosis* (ATCC22019), *Haemophilus influenzae* (ATCC49247), and *Streptococcus pneumoniae* (ATCC49619), were sourced from Wenzhou Kontai.

## Data analysis

Susceptibility outcomes were systematically analyzed using WHONET 5.6 software. Statistical inferences were made using SPSS 25. Categorical data were summarized as frequencies (*n*) and percentages (%), with the significance assessed using Fisher's exact test. A non-parametric Wilcoxon signed-rank test was used to analyze non-normally distributed continuous variables, with statistical significance set at $P < 0.05$. Trends in antibiotic resistance were identified using Cochran–Armitage tests.

## RESULTS

### Patient demographics and MK culture positivity

A comprehensive analysis of the samples of 3,691 patients with MK revealed a positivity rate of 38.72%, comprising 1,429 positive cases. Following rigorous deduplication, 1,382 unique cases were subjected to further analysis, showing a higher proportion of men, with a sex ratio of 2.43:1 (979 men vs. 403 women). Patient ages ranged from 2 to 93 years, with a mean age of 58.6 ± 14.7 years. Notably, 1,315 patients harbored a single pathogen strain; 55 patients harbored two strains; and 12 patients harbored three or more strains, collectively contributing to 1,469 strains for detailed pathogen spectrum and resistance profiling.

### Pathogen composition analysis

Of the 1,469 strains, 40.78% (*n* = 599) were bacterial, comprising 23.14% (*n* = 340) gram-positive and 17.63% (*n* = 259) gram-negative strains. The predominant bacterial pathogens were *P. aeruginosa* (4.97%), *S. pneumoniae* (4.36%), *Staphylococcus epidermidis* (4.29%), *S. aureus* (2.93%), *Corynebacterium macginleyi* (0.75%), and *Mycobacterium abscessus* (0.75%). In contrast, fungi constituted 58.34% (*n* = 857) of cases, with primarily filamentous fungi comprising 51.94% (*n* = 763) and yeast-like fungi constituting 6.40% (*n* = 94), yielding a significant ratio of 8.52:1. Key filamentous fungi included *Fusarium* spp. (21.38%), *Aspergillus* spp. (6.60%), and *Alternaria* spp. (4.36%), whereas predominant yeast-like fungi included *C. parapsilosis* (3.27%) and *Candida albicans* (0.88%). Additionally, *Acanthamoeba* was isolated from 13 cases (0.88%), underscoring its clinical relevance. A summary of the microbiological profiles of MK in Southern Zhejiang over the 8-year study period is depicted in Fig. 1. The composition of isolated strains

over the 8-year study period is depicted in Fig. 2. Among the 67 mixed infections, 36 were bacterial-only mixed infections, involving *Streptococcus* spp., *Staphylococcus* spp., *Corynebacterium* spp., *Haemophilus* spp., and *Chryseobacterium* spp. Another 26 were bacterial–fungal mixed infections predominantly featuring *Staphylococcus* spp., *Fusarium* spp., and *Candida* spp. The remaining five cases were fungal-only mixed infections primarily involving *Candida* spp. combined with filamentous fungi. For detailed information, refer to Table 1.

## Causative factors

Trauma and foreign body entry emerged as the primary precipitants of bacterial, fungal, and *Acanthamoeba* keratitis, accounting for 52.17% (721/1,382) of cases. Plant-associated injuries predominated, comprising 22.21% of the total injuries. Postoperative infections comprised 8.03% of cases, whereas contact lens-related factors accounted for a minor 1.23%. Notably, the etiology remained elusive in 34.15% of cases, highlighting the need for further investigation. Comprehensive data on the predisposing factors of patients with MK are presented in Table 2.

## Antimicrobial resistance profile

Gram-positive bacteria exhibited significant resistance patterns, particularly toward erythromycin (71.7%), penicillin (58.5%), ofloxacin (53.3%), levofloxacin (30.9%), gentamicin (50.0%), and vancomycin, albeit at a lower rate of 2.3%. In contrast, gram-negative bacteria exhibited resistance to piperacillin/tazobactam (14.0%), levofloxacin (10.5%), gentamicin (17.4%), ceftazidime (11.8%), and meropenem (4.2%). Specifically, *S. epidermidis* exhibited alarmingly high resistance to erythromycin (67.4%), penicillin

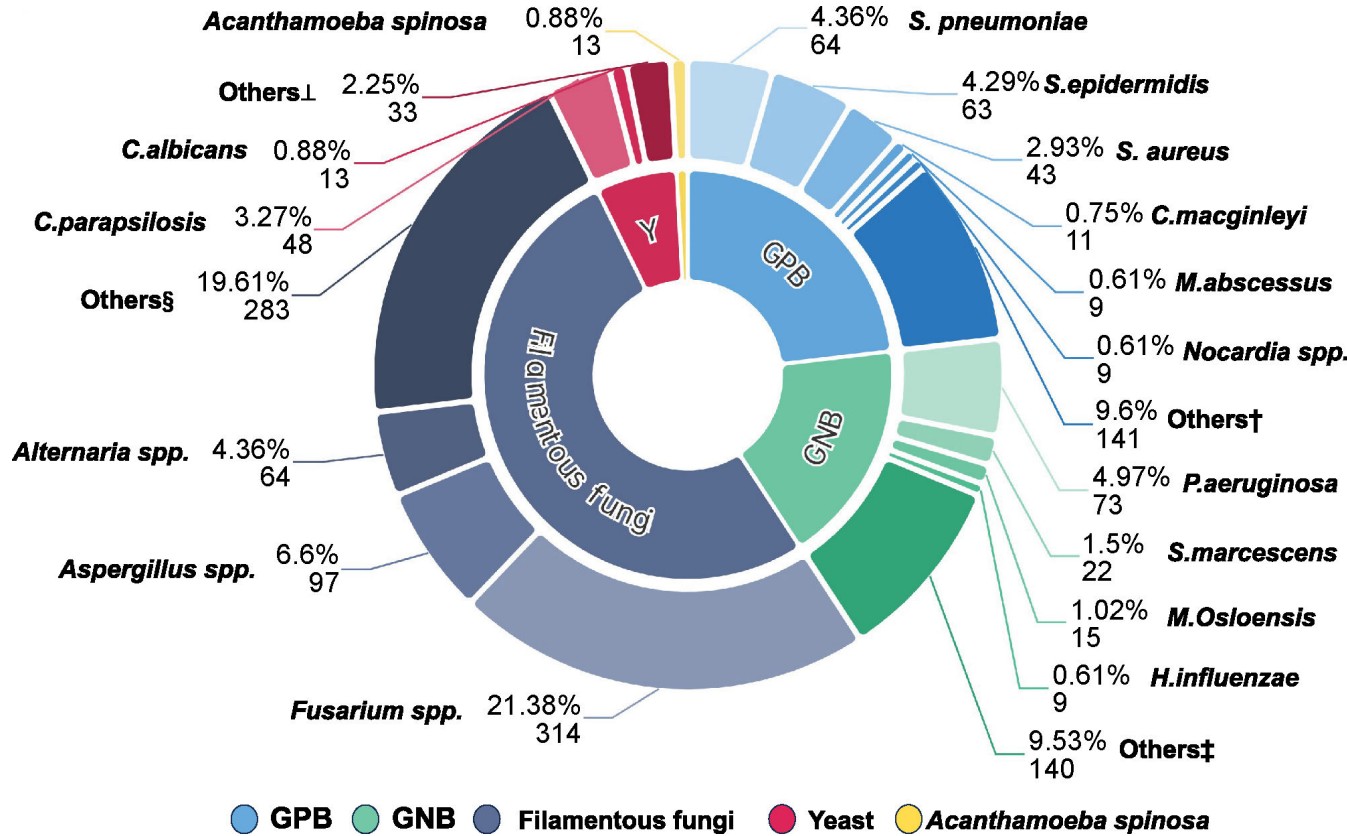

**FIG 1**  Summary of microbiological profiles of microbial keratitis in Southern Zhejiang Region over an 8-year period. GPB, gram-positive bacteria, GNB, gram-negative bacteria. Others† include *S. haemolyticus*, *C. macillonii*, *S. oralis*, etc. Others‡ include *H. influenzae*, *K. pneumoniae*, *S. maltophilia*, *A. baumannii*, etc. Others§ include *Curvularia* spp., *Colletotrichum* spp., *Paecilomyces* spp., etc. Others⊥ include *Candida tropicalis*, *Candida guilliermondii*, etc.

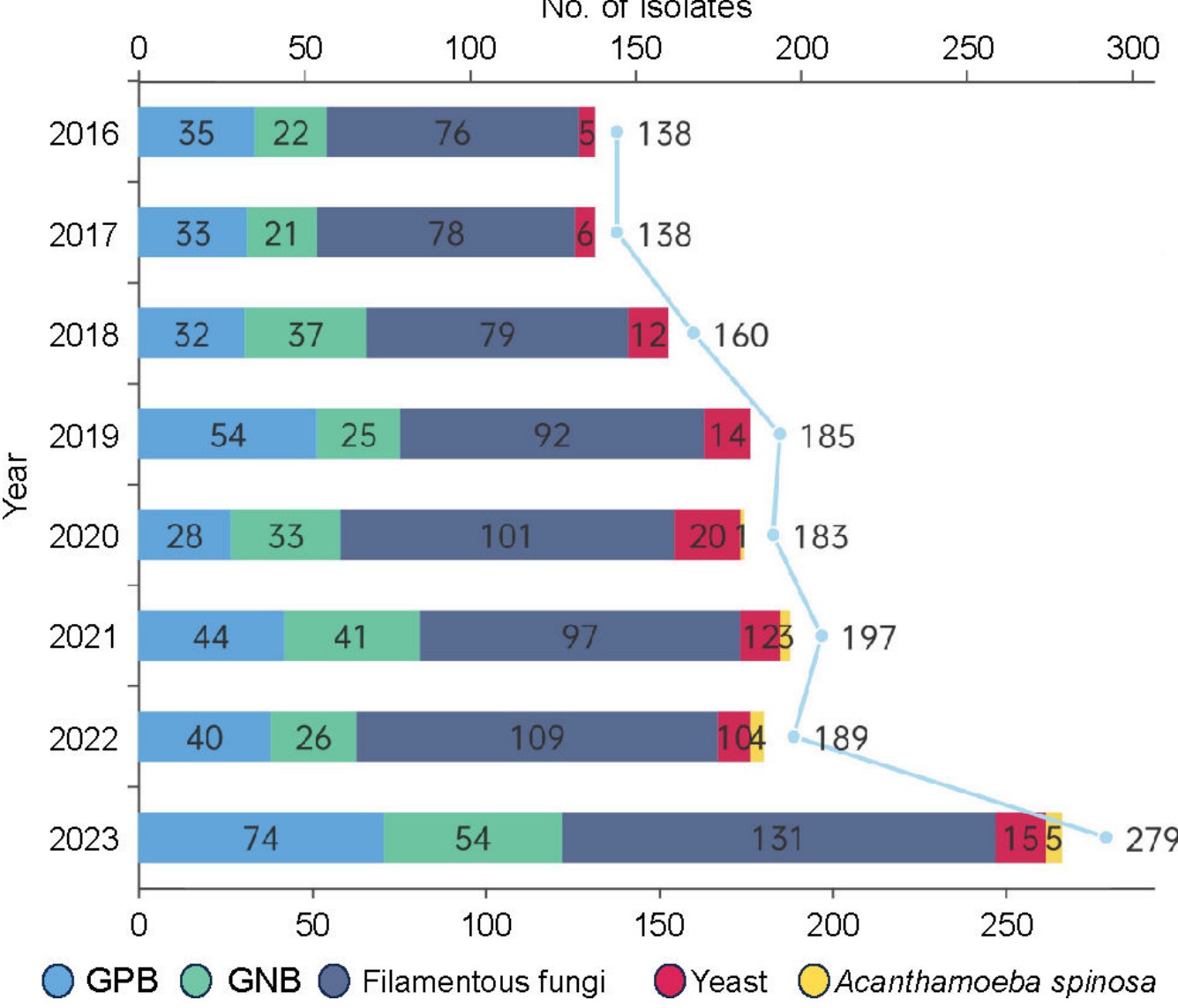

**FIG 2** Composition of isolated strains over an 8-year period (*P* < 0.05; Fisher's exact test).

(98.2%), oxacillin (57.1%), levofloxacin (54.0%), and gentamicin (37.0%) but remained fully susceptible to vancomycin. Notably, *S. aureus* demonstrated marked multidrug resistance to penicillin, tobramycin, and erythromycin, whereas *S. pneumoniae* demonstrated marked resistance to erythromycin. Comprehensive data on the MICs and resistance profiles for corneal bacterial isolates from 2016 to 2023 are presented in Table 3.

The rates of *C. albicans* and *C. parapsilosis* resistance to voriconazole were 46.2 (6/13) and 0% (0/48), respectively. The rates of *Aspergillus fumigatus* resistance to voriconazole, itraconazole, and amphotericin B were 2.9 (1/35), 32.4, and 42.9%, respectively. The rate of *Aspergillus flavus* resistance to itraconazole was 16.3%, with $MIC_{50}$ values of 0.125 and 1.5 µg/mL for voriconazole and amphotericin B, respectively. For *Fusarium* spp., the $MIC_{50}$ values against voriconazole, itraconazole, and amphotericin B were 1, 64, and 1 µg/mL, respectively. For other filamentous fungi, the $MIC_{50}$ values against voriconazole, itraconazole, and amphotericin B were 0.19, 0.75, and 0.5 µg/mL, respectively. Comprehensive data on MICs and resistance profiles for corneal fungal isolates from 2016 to 2023 are outlined in Table 4.

TABLE 1 Composition of 67 cases of mixed infection[a]

| Category | Bacterial-only mixed infection, n (%) (N = 36) | Bacterial–fungal mixed infection, n (%) (N = 26) | Fungal-only mixed infection, n (%) (N = 5) |
|---|---|---|---|
| Main isolate | *Streptococcus* spp. 15 (41.67) | *Staphylococcus* spp. 12 (46.15) | *Candida* spp. 5 (100.00) |
| | *Staphylococcus* spp. 8 (22.22) | *Fusarium* spp. 9 (34.62) | Filamentous fungi 4 (80.00) |
| | *Corynebacterium* spp. 7 (26.92) | *Candida* spp. 5 (19.23) | |
| | *Haemophilus* spp. 6 (23.08) | | |
| | *Chryseobacterium* spp. 5 (19.23) | | |

[a]n, individuals infected by a specific microbe; N, total individuals with mixed infections.

## Resistance trends and multidrug resistance

The resistance rates of corneal isolates to common antimicrobial agents, including quinolones, aminoglycosides, and β-lactams, were analyzed over an 8-year period to observe trends in antimicrobial resistance (Fig. 3 and 4). From 2016 to 2023, a slight increase was observed in resistance to ceftazidime ($P = 0.002$), meropenem ($P = 0.005$), and erythromycin ($P = 0.0291$; Fig. 3). Gram-negative bacteria showed a significant increase in resistance to tobramycin ($P = 0.002$) and ceftazidime ($P < 0.001$), along with a decrease in resistance to gentamicin ($P < 0.001$; Fig. 3). *S. pneumoniae* demonstrated a significant decrease in resistance to penicillin ($P < 0.001$; Fig. 4), whereas *P. aeruginosa* exhibited a significant increase in resistance to ceftazidime ($P < 0.001$), gentamicin ($P < 0.001$), and tobramycin ($P = 0.002$; Fig. 4).

The surveillance of multidrug-resistant bacteria revealed that methicillin-resistant *Staphylococcus* (MRS) was the most prevalent, with 66 isolates identified, including 36 *S. epidermidis* and nine *S. aureus* strains. The detection rate of methicillin-resistant coagulase-negative *Staphylococcus* (MRScon) remained high (50.0–90.9%); however, no significant trend was observed ($P = 1.405$; Fig. 4). The annual detection rate of methicillin-resistant *S. aureus* (MRSA) varied, peaking at 60.0% in 2017 but with no detection in 2018, 2020, and 2023 (Fig. 4). The detection rate of penicillin-resistant *S. pneumoniae* decreased over most years, with an increase observed in 2021 ($P < 0.001$), maintaining a

TABLE 2 Predisposing factors of patients with microbial keratitis

| Predisposing factors | Number of patients, n (%) (N = 1382) | Gram-positive Isolates, n (%) (N = 340) | Gram-negative Isolates, n (%) (N = 259) | Filamentous isolates, n (%) (N = 763) | Yeast isolates, n (%) (N = 94) | *Acanthamoeba spinosa* isolates, n (%) (N = 13) |
|---|---|---|---|---|---|---|
| Traumatic | 721 (52.17) | 130 (38.24) | 114 (44.02) | 470 (61.60) | 27 (28.72) | 5 (38.46) |
| Plant-related | 307 (22.21) | 38 (11.18) | 32 (12.36) | 235 (30.80) | 9 (9.57) | 1 (7.69) |
| Metal-related | 92 (6.66) | 17 (5.00) | 25 (9.65) | 48 (6.29) | 5 (5.32) | 0 (0) |
| Hand-related | 27 (1.95) | 7 (2.06) | 8 (3.09) | 12 (1.57) | 1 (1.06) | 1 (7.69) |
| Others[a] | 295 (21.35) | 68 (20.00) | 49 (18.92) | 175 (22.94) | 12 (12.77) | 3 (23.08) |
| Ophthalmic surgeries | 111 (8.03) | 36 (10.59) | 20 (7.72) | 25 (3.28) | 37 (39.36) | 0 (0) |
| Post-keratoplasty | 68 (4.92) | 18 (5.29) | 12 (4.63) | 15 (1.97) | 28 (29.79) | 0 (0) |
| Others[b] | 43 (3.11) | 18 (5.29) | 8 (3.09) | 10 (1.31) | 9 (9.57) | 0 (0) |
| Ocular diseases[c] | 61 (4.41) | 24 (7.06) | 10 (3.86) | 23 (3.01) | 8 (8.51) | 0 (0) |
| Contact lens-related | 17 (1.23) | 8 (3.37) | 7 (2.70) | 2 (0.26) | 0 (0) | 2 (14.94) |
| Unknown | 472 (34.15) | 142 (41.76) | 108 (41.70) | 243 (31.85) | 22 (23.40) | 6 (44.81) |

[a]Sand, dust, insects, chemicals, etc.
[b]Cataract surgery, penetrating keratoplasty, pterygium excision, etc.
[c]Exposure keratitis, corneal dystrophy, etc.

**TABLE 3** Minimum inhibitory concentrations and resistance profiles for corneal bacterial isolates from 2016 to 2023[a]

| Antimicrobials | Gram-positive bacteria ($n = 340$) | S. epidermidis ($n = 63$) | S. aureus ($n = 43$) | S. pneumonia ($n = 64$) | Gram-negative bacteria ($n = 259$) | S. aeruginosa ($n = 73$) |
|---|---|---|---|---|---|---|
| Penicillin G, R* ($MIC_{50}$) | 58.5 (1) | 98.2 (8) | 91.7 (8) | 4.16 (1) | N/A | N/A |
| Piperacillin–tazobactam, R ($MIC_{50}$) | N/A | N/A | N/A | N/A | 14.0 (8/4) | 8.6 (8/4) |
| Oxacillin, R ($MIC_{50}$) | 45.5 (1) | 57.1 (1) | 20.9 (0.5) | N/A | N/A | N/A |
| Ceftriaxone, R ($MIC_{50}$) | 5.8 (0.5) | N/A | N/A | 0.0 (0.5) | 20.0 (0.25) | N/A |
| Ceftazidime, R ($MIC_{50}$) | N/A | N/A | N/A | N/A | 11.8 (2) | 4.4 (2) |
| Meropenem, R ($MIC_{50}$) | N/A | N/A | N/A | N/A | 4.2 (0.25) | 0.0 (0.5) |
| Erythromycin, R ($MIC_{50}$) | 71.7 (16) | 67.4 (16) | 62.1 (8) | 90.5 (32) | N/A | N/A |
| Cotrimoxazole, R ($MIC_{50}$) | 23.4 (0.25/4.75) | N/A | N/A | N/A | 44.4 (2/38) | N/A |
| Ofloxacin, R ($MIC_{50}$) | 53.3 (8) | 69.5 (8) | 38.5 (1) | 11.1 (1) | 14.6 (1) | 11.5 (1) |
| Levofloxacin, R ($MIC_{50}$) | 30.9 (1) | 54.0 (4) | 30.6 (0.5) | 0.0 (1) | 10.5 (0.5) | 4.8 (0.5) |
| Gatifloxacin, R ($MIC_{50}$) | 24.0 (0.5) | 38.5 (1) | N/A | 0.0 (0.25) | 12.0 (0.5) | N/A |
| Clindamycin, R ($MIC_{50}$) | 53.7 (2) | N/A | N/A | N/A | N/A | N/A |
| Gentamicin, R ($MIC_{50}$) | 50.0 (4) | 37.0 (8) | 34.4 (1) | N/A | 17.4 (1) | 6.8 (2) |
| Amikacin, R ($MIC_{50}$) | 8.3 (4) | N/A | N/A | N/A | 11.7 (4) | 1.8 (4) |
| Tobramycin, R ($MIC_{50}$) | 46.8 (16) | 47.6 (8) | 42.9 (8) | N/A | 21.1 (2) | 6.9 (1) |
| Chloramphenicol, R ($MIC_{50}$) | 11.4 (4) | 24.6 (8) | 9.1 (8) | 6.7 (2) | 29.8 (8) | 0.0 (32) |
| Vancomycin, R ($MIC_{50}$) | 2.3 (1) | 0.0 (2) | 0.0 (1) | 0.0 (2) | N/A | N/A |
| Linezolid, R ($MIC_{50}$) | 0.0 (1) | 0.0 (1) | 0.0 (1) | N/A | N/A | N/A |
| Tetracycline, R ($MIC_{50}$) | 45.0 (4) | 31.3 (4) | 22.7 (0.5) | 70.5 (16) | 50.0 (1) | N/A |

[a]R*, rate of drug resistance (%); N/A, not applicable; $MIC_{50}$, the minimum inhibitory concentration (µg/mL) that inhibited 50% of the tested microorganisms.

low average level of 4.16% (0.0–33.3%; Fig. 4). Carbapenem-resistant Enterobacteriaceae, carbapenem-resistant *P. aeruginosa*, and vancomycin-resistant *Enterococcus* were not detected (Fig. 4). Additionally, during the study period, four cases of pan-drug-resistant *Chryseobacterium meningosepticum* and six cases of pan-drug-resistant *M. abscessus* were identified.

## In vitro resistance rates based on patient age

In keratitis caused by *Staphylococcus* spp., a general pattern of an increasing mean percentage of *in vitro* antibiotic resistance was observed with each advancing decade of life ($P < 0.001$; Fig. 5). Pairwise comparisons revealed notable differences between isolates from younger and older patients, indicating an age-related increase in antibiotic resistance. In contrast, no correlation was observed between age and the mean percentage of resistance in *P. aeruginosa* and *S. pneumoniae*.

## Consistency analysis of quinolone drugs

A comprehensive statistical analysis was performed on quinolone drugs commonly used for ocular applications. The results indicated a high level of consistency in susceptibility test outcomes between ofloxacin and levofloxacin at 85.7% ($n = 154$; Fig. 6), with a very major discrepancy (VMD) rate of 3.8%. Similarly, the consistency of the susceptibility test outcomes between levofloxacin and gatifloxacin was 82.0% ($n = 172$; Fig. 6), with a VMD rate of 5.2%. Overall, the *in vitro* susceptibility test results for quinolone drugs demonstrated a notable degree of consistency.

## Summary

Our study identified trauma and the introduction of foreign bodies as key predisposing factors for MK in our region, with *Fusarium* being the primary causative agent. Mixed infections pose significant challenges to the clinical management of MK. Notably, gram-positive bacteria, particularly *Staphylococcus* spp., exhibit high resistance to

**TABLE 4** Minimum inhibitory concentrations and resistance profiles for corneal fungal isolates from 2016 to 2023[a]

| Antimicrobials | | C. albicans $n = 13$ | C. parapsilosis $n = 48$ | Other yeast $n = 33$ | P-value | Fusarium spp. $n = 314$ | A. flavus $n = 49$ | A. fumigatus $n = 35$ | Other filamentous fungi $n = 365$ | P-value |
|---|---|---|---|---|---|---|---|---|---|---|
| Voriconazole | R* (%) | 46.2 (6/13) | 0 (0/48) | 5.9 | **0.043** | N/A | N/A | 2.9 (1/35) | N/A | **<0.001** |
| | $MIC_{50}$ (µg/mL) | 0.19 | 0.016 | 0.032 | | 1 | 0.125 | 0.125 | 0.19 | |
| | $MIC_{90}$ (µg/mL) | 64 | 0.064 | 0.75 | | 8 | 0.5 | 0.38 | 1.5 | |
| Itraconazole | R (%) | N/A | N/A | N/A | **0.043** | N/A | 16.3 | 32.4 (11/34) | N/A | **<0.001** |
| | $MIC_{50}$ (µg/mL) | 0.19 | 0.19 | 0.25 | | 64 | 0.5 | 0.75 | 0.75 | |
| | $MIC_{90}$ (µg/mL) | 32 | 0.25 | 1.5 | | 64 | 4 | 4 | 64 | |
| Amphotericin | R (%) | N/A | N/A | N/A | **0.049** | N/A | N/A | 42.9 (15/35) | N/A | **<0.001** |
| | $MIC_{50}$ (µg/mL) | 0.25 | 0.25 | 0.75 | | 1 | 1.5 | 1 | 0.5 | |
| | $MIC_{90}$ (µg/mL) | 0.75 | 1 | 64 | | 64 | 32 | 8 | 64 | |

[a]R*, rate of drug resistance; N/A, not applicable; $MIC_{50}$ and $MIC_{90}$ are defined as the minimum inhibitory concentrations that inhibited 50 and 90% of the tested microorganisms, respectively; significant P-values are indicated in bold.

quinolones, aminoglycosides, and β-lactams, with resistance patterns correlating with patient age. Continuous monitoring has revealed a slight increase in resistance to cephalosporins, including ceftazidime, meropenem, and erythromycin. The potential for fungal resistance in the treatment of fungal keratitis warrants clinical attention.

## DISCUSSION

In this study, the culture positivity rate for MK corneal scrapings was 38.72%, consistent with the findings of some previous studies (8). However, other studies have reported higher culture positivity rates, with a median rate of 50.3% (range: 32.6–79.4%) in clinically diagnosed MK cases (5). The relatively lower culture positivity rate observed in our study may be attributed to the prior use of antibiotics by patients before seeking medical attention (9), inadequate sampling from infected corneas, or sampling in non-infectious cases, such as sterile corneal melts and marginal keratitis. Additionally, infections caused by viruses or anaerobic bacteria in some patients may lead to challenges in obtaining pathogenic bacteria through routine cultures. For patients with a history of antibiotic use before visiting the hospital, our standard practice is to discontinue all antibiotics for 24–48 h before performing corneal scraping to reduce the effect of antibiotics on the culture results.

Common risk factors for MK include contact lens wear, ocular trauma, ocular surface disease, diabetes, and ocular surgery (10, 11), consistent with our observations. In the present study, more than half (52.17%) of the patients with positive cultures had trauma or foreign body entry as the primary cause of MK. Additionally, a notable imbalance in the male-to-female ratio (2.43:1) was observed, likely related to the high risk of eye injuries among individuals engaged in heavy physical work, such as construction and agriculture, where the workforce is predominantly men. A recent study corroborates our findings, indicating that ocular trauma in men is the most important risk factor for MK occurrence (12). A small percentage (1.95%) of patients reported that their symptoms began after rubbing their eyes, warranting further consideration. This occurrence could be attributed to improper handling after a foreign body enters the eye, exacerbating corneal injury, or a lack of attention to hand hygiene, causing corneal infection. Therefore, standardized management of ocular diseases caused by trauma or foreign body entry is crucial to preventing secondary MK. Successful public health interventions

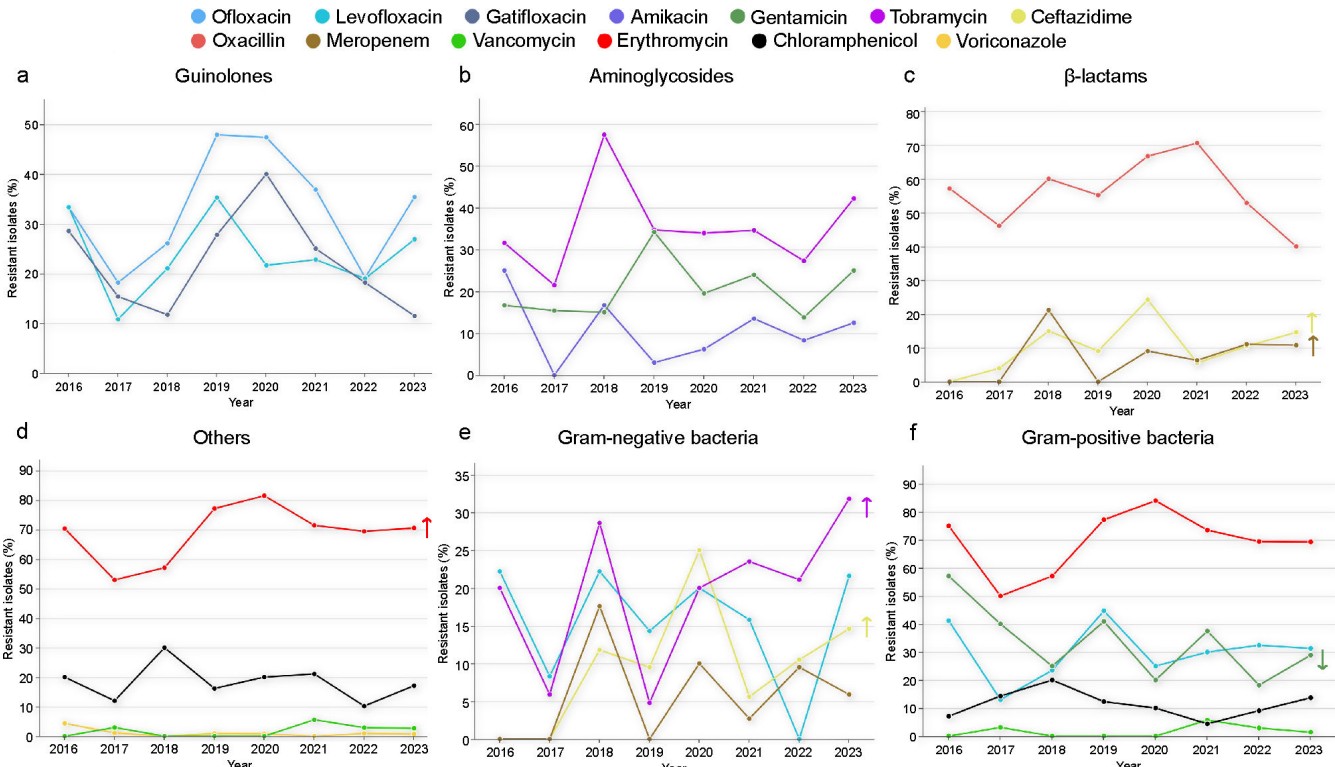

**FIG 3** (a–d) Resistance trends among quinolones, aminoglycosides, β-lactams, and other antibiotics over an 8-year period. (e–f) Resistance trends among gram-positive and -negative bacteria over the 8-year study period. Cochran–Armitage tests were used to identify significant decreasing (↓) and increasing (↑) trends in antibiotic resistance.

in Bhutan (13) and Nepal (14) have provided promising strategies for preventing MK. These interventions include training primary health workers and volunteers in ocular anatomy and basic eye care, such as the use of fluorescein sodium test paper and penlights to identify corneal abrasions and early prophylactic use of antibiotics (within 6 h) for patients meeting the criteria for corneal abrasions. Early treatment reduces the risk of corneal scarring, vascularization, or perforation, which could otherwise necessitate corneal transplants. Additionally, a study in the United States demonstrated an association between the place of residence of patients and the severity of visual impairment at the time of presentation (15). Therefore, enhancing the training of primary medical institutions in ocular trauma diagnosis and treatment, along with community education in our region of study, could be a viable approach to preserving visual health in patients.

Infections secondary to ophthalmic surgeries, including cataract surgery with intraocular lens implantation, glaucoma drainage valve implantation, laser correction for myopia, intraocular injection for uveitis, and post-keratoplasty, accounted for 8.03% of cases. Among these, post-keratoplasty was the most predominant surgical factor, comprising 4.92% of cases. Notably, 29.79% of yeast-like fungal infections were isolated from post-keratoplasty procedures. This high rate of infection may be attributed to the susceptibility of the transplanted cornea to further microbial invasion (16). Contact lens-related diseases accounted for 1.23% of cases, representing an important risk factor for *Acanthamoeba* keratitis. However, the etiology of 34.15% of cases remained unknown, indicating that the pathogenesis of a considerable proportion of MK in the study region remains unclear.

Furthermore, fungi (58.34%), particularly filamentous fungi (51.94%), constituted the primary corneal pathogens in this region. Trauma, particularly plant-related injuries (30.80%), was the most common cause of filamentous fungal keratitis. The predominant

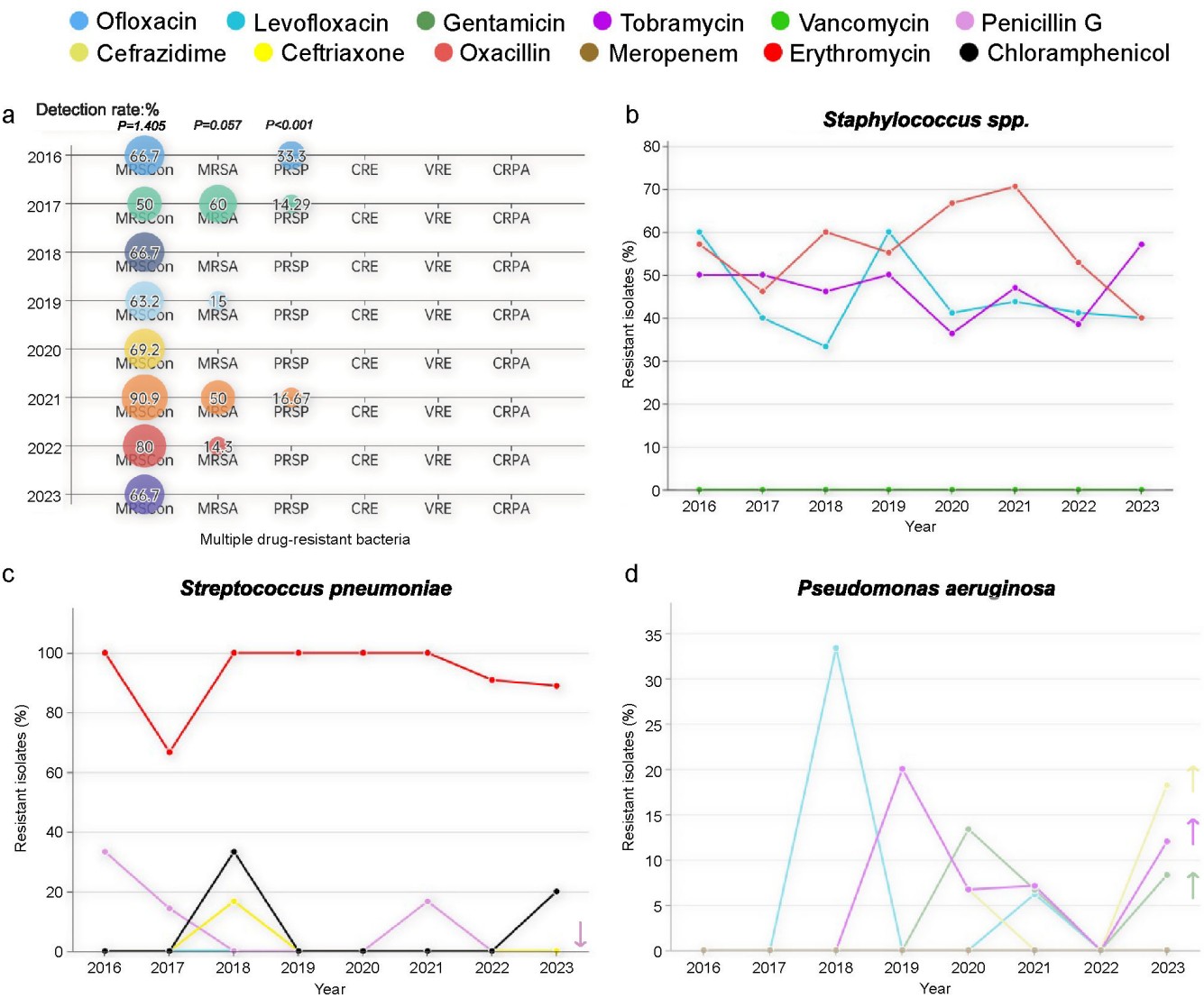

**FIG 4** (a) Multidrug-resistant bacterial isolates from patients with microbial keratitis over an 8-year study period. (b–d) Resistance trends among *Staphylococcus* spp., *Pseudomonas aeruginosa*, and *Streptococcus pneumoniae* isolates from patients with microbial keratitis over the study period. Cochran–Armitage tests were used to identify significant decreasing (↓) and increasing (↑) trends in antibiotic resistance.

filamentous fungi included *Fusarium*, *Aspergillus*, and *Alternaria*, with *Fusarium* spp. being the most frequently isolated fungi from the cornea (21.38%), significantly higher than that of the second-ranked *Aspergillus* (6.60%). Among *Candida* spp., *C. parapsilosis* and *C. albicans* were the most common, with *C. parapsilosis* having a higher isolation rate (3.27%) than *C. albicans* (0.88%). A systematic review of 36 studies revealed that bacteria are the most commonly isolated pathogens in infectious keratitis in developed countries, whereas fungi are more prevalent in developing countries (17). In the United States, approximately 6% of suspected infectious keratitis cases are caused by fungi, with well-documented risk factors, such as contact lens use, ocular trauma, and local corticosteroids (18). The Infectious Keratitis Study conducted by the Asia Cornea Society demonstrated that fungi are the most common pathogens in China (19). This observation is likely attributed to the low level of mechanization in agricultural activities in this region, necessitating farmers to participate in extensive manual labor, which increases the probability of plant-related corneal injuries. Statistical data further indicate that plant injury is the most important risk factor for filamentous fungal keratitis. Fungal keratitis is a serious blinding ocular disease, and the incorrect use of corticosteroids when the cause

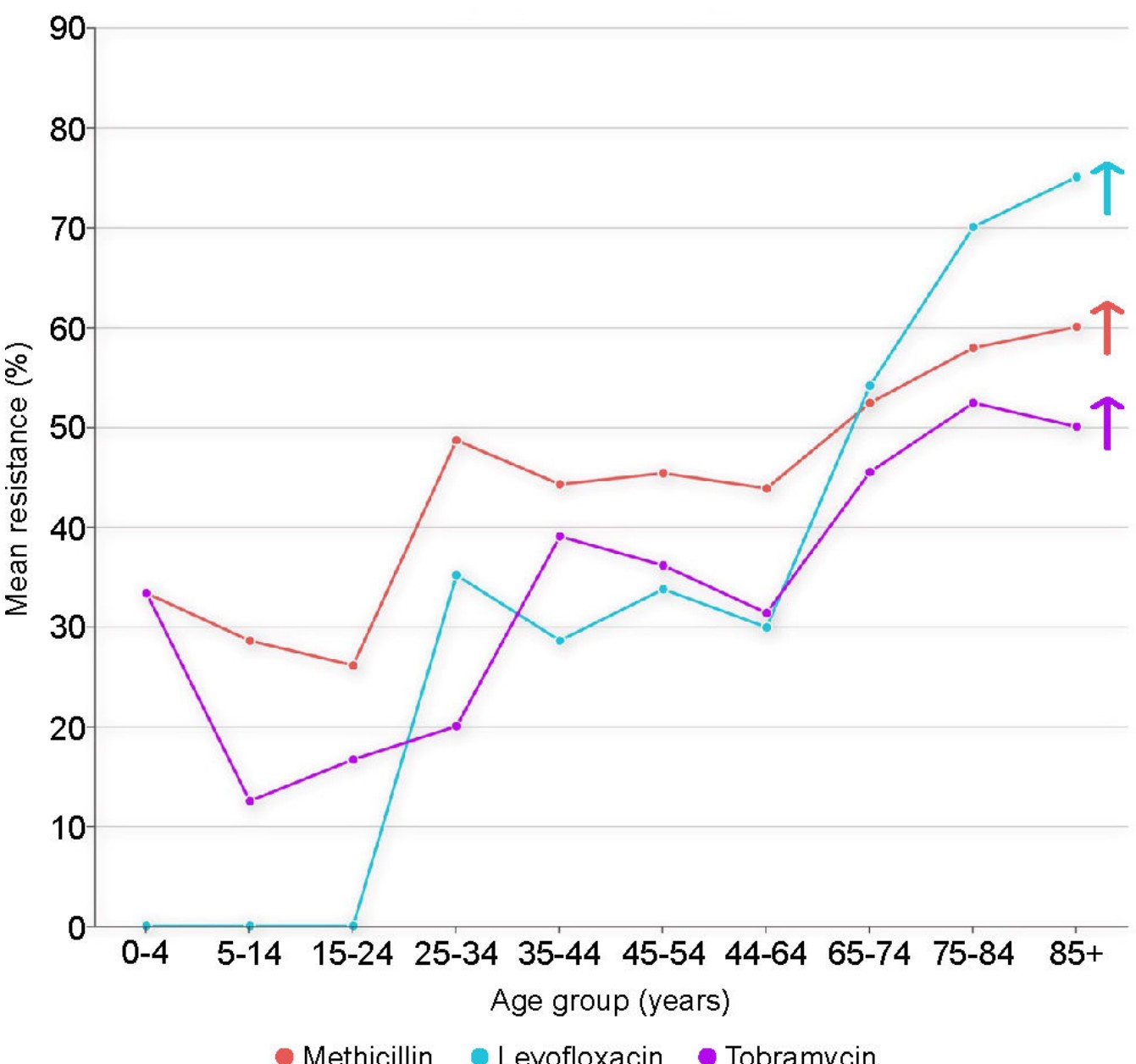

**FIG 5** Resistance in staphylococcal isolates from the cornea, stratified by patient age. Cochran–Armitage tests were used to identify significant decreasing (↓) and increasing (↑) trends in antibiotic resistance.

is unclear can lead to uncontrolled infection and eventual blindness, posing a significant challenge to medical professionals in this region.

Global cases of *Acanthamoeba* keratitis account for approximately 1–3% of all MK cases (5), although the monitoring results in the present study were slightly lower at 0.88%. This discrepancy may be attributed to the fact that *Acanthamoeba* culture was only initiated in 2020. Calculating from 2020 onwards, the positive rate increased to 1.53% (13/848). *Acanthamoeba* keratitis has a poor prognosis; however, accurate and timely diagnosis can facilitate vision preservation in patients. *Acanthamoeba* culture is a simple and feasible laboratory diagnostic method, with our current culture-positive

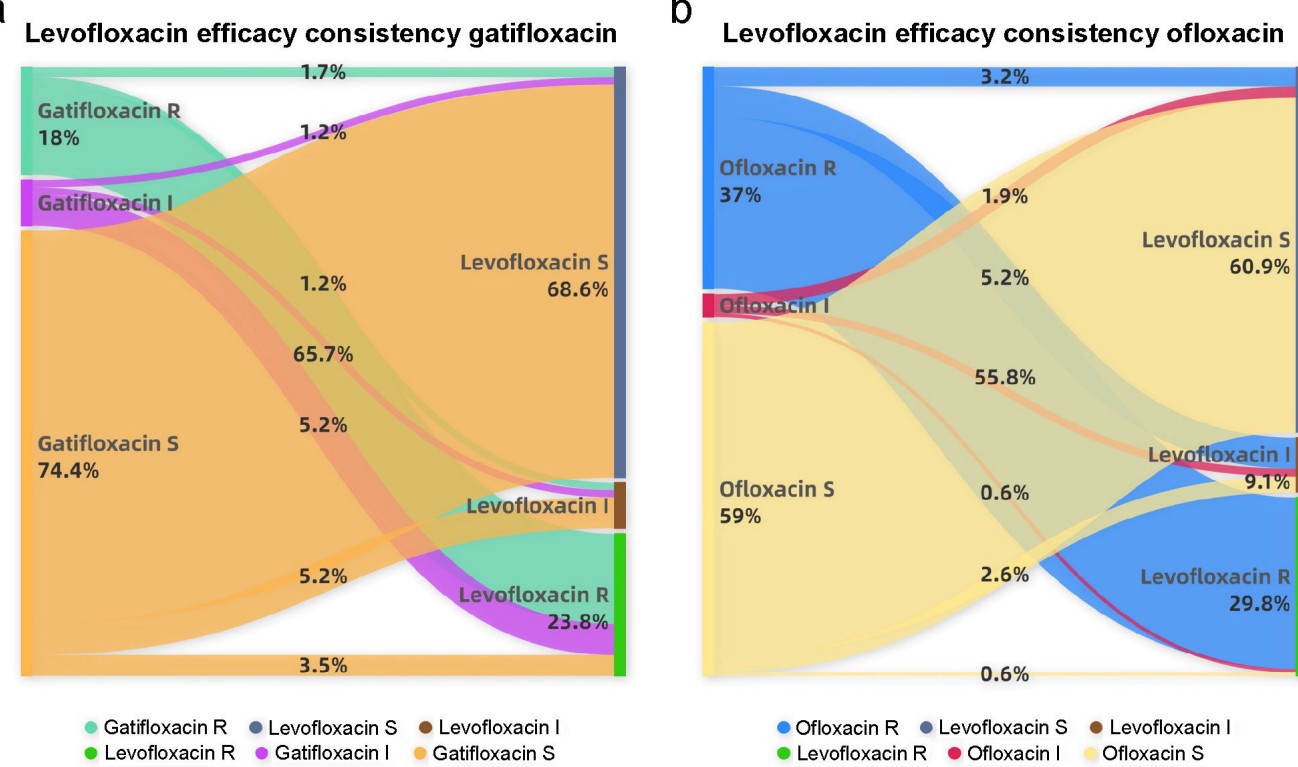

**FIG 6** Consistency analysis of the sensitivity results of quinolone drugs. R, rate of drug resistance; I, rate of drug intermediates; and S, rate of drug sensitivity.

rate approaching 100%. This method can be widely adopted in ophthalmic specialist hospitals.

Our study revealed that the proportion of gram-positive bacteria (*n* = 340, 56.76%) was slightly higher than that of gram-negative bacteria (*n* = 259, 43.24%) among the bacterial isolates. The highest isolation rates were observed for *P. aeruginosa*, *S. pneumoniae*, *S. epidermidis*, and *S. aureus*. Despite local and regional variations in the etiology of bacterial keratitis, the most commonly reported pathogenic microorganisms appear to be consistent worldwide, with statistics showing a higher proportion of gram-positive bacteria than gram-negative bacteria (8). However, these numbers should be interpreted cautiously because most commensal bacteria on the eyelid and ocular surface are gram-positive and may contaminate samples. To address this issue, we performed Gram staining microscopy of corneal scraping smears in conjunction with cultures. This approach offers several advantages: it can rapidly guide clinical medication use and exclude contaminating bacteria, thereby improving the detection rate of pathogenic bacteria (20). Notably, 4.85% of cases involved cultures with two or more types of bacteria, with some cases showing mixed fungal and bacterial infections. This increases the complexity of clinical management and highlights the importance of microbial cultivation and submission for diagnostic testing. Smear examinations can verify the accuracy of these culture results and guide medication, particularly in cases where antibacterial agents inhibit growth or culture conditions are not optimal, making it a necessary auxiliary tool for the diagnosis and treatment of MK.

Antibacterial susceptibility testing results revealed that the resistance rates of gram-positive bacteria to penicillin G, erythromycin, clindamycin, gentamicin, and ofloxacin exceeded 50%. Similarly, the resistance rates of gram-negative bacteria to tetracycline exceeded 50%. Overall, gram-positive bacteria exhibited more severe resistance to antibacterial agents than gram-negative bacteria. For example, the resistance rate of gram-positive bacteria to the broad-spectrum antibacterial agent ofloxacin reached 53.3%, whereas that of gram-negative bacteria was 14.6%. The

resistance rate of gram-positive bacteria to tobramycin reached 46.8%, whereas that of gram-negative bacteria was 21.1%. However, except for *Mycobacterium tuberculosis* and *Nocardia*, no gram-positive bacteria resistant to vancomycin were detected. Furthermore, none of the gram-positive strains tested for linezolid susceptibility showed resistance. Therefore, vancomycin and linezolid remain effective drugs against gram-positive bacteria.

Antifungal susceptibility testing results revealed that corneal *C. albicans* isolates exhibited a high resistance rate of 46.2% to voriconazole, significantly higher than the 0% resistance rate observed for *C. parapsilosis* ($P < 0.001$). Given that voriconazole is one of the few and widely used antifungal agents in ocular treatment, accurate identification of *Candida* spp. and susceptibility testing are particularly crucial for effectively preventing treatment failures. Although clinical breakpoints for filamentous fungi have not been established, minimum effective concentration values (ECVs) have been defined for certain *Aspergillus* species and species complexes against amphotericin B, caspofungin, itraconazole, isavuconazonium, posaconazole, and voriconazole. These ECVs facilitate the identification of non-wild-type *Aspergillus* species and species complexes with potential resistance. In our study, the $MIC_{90}$ of voriconazole against isolated *A. flavus* was 0.5 µg/mL, which was lower than its ECV of 2 µg/mL, indicating that most clinically isolated *A. flavus* strains do not exhibit potential resistance to voriconazole. However, the $MIC_{90}$ values for itraconazole and amphotericin B against *A. flavus* were 4 and 32 µg/mL, respectively, both exceeding their corresponding ECVs of 1 and 4 µg/mL, respectively, suggesting potential resistance in some strains. Further statistical analysis revealed significant differences in the *in vitro* susceptibility of different corneal filamentous fungi isolates to voriconazole, itraconazole, and amphotericin B. *Fusarium* spp. exhibited varying degrees of resistance to multiple antifungal agents. Therefore, quantitative *in vitro* susceptibility testing of filamentous fungi is particularly urgent in regions where filamentous fungi are the major components of ocular isolates. This approach not only provides essential guidance for clinical drug use but also holds importance for accumulating epidemiological data.

Furthermore, statistical analysis of resistance trends from 2016 to 2023 revealed that the overall resistance of corneal isolates in our region of study remained stable. Specifically, resistance to ceftazidime, meropenem, and erythromycin showed a slight upward trend. A slight decrease in resistance to gentamicin was observed exclusively in gram-positive bacteria. Notably, resistance levels for erythromycin, tobramycin, and ofloxacin are already high, with ophthalmic formulations of these drugs readily available from non-hospital pharmacies. Therefore, the standardization and rational use of ocular antibacterial agents to delay further increases in bacterial resistance are imperative. Additionally, our results demonstrated good consistency in *in vitro* susceptibility testing across different generations of quinolones. Therefore, the cautious and rational use of antibacterial agents is essential for preserving their clinical efficacy.

MRS, with a detection rate as high as 45.5%, was the most prevalent multidrug-resistant bacterium isolated from corneal samples in the region of study. Specifically, the detection rate of methicillin-resistant *S. epidermidis* was 57.1%, whereas that of MRSA was 20.9%. No significant changes were observed in the detection rates of MRScon and MRSA. This trend contrasts with the overall bacterial resistance trend in China, where the isolation rates of major multidrug-resistant bacteria, including MRS, have shown a slight decline since 2016 (21). Globally, significant regional differences exist in the prevalence of multidrug-resistant bacteria isolated from the cornea. For example, in the United Kingdom, the detection rate of MRSA ranges from 0.1 to 5.0% (8). In the United States, the Antibiotic Resistance Monitoring in Ocular Microbes Study reported detection rates of 41.9 and 34.5% for MRScon and MRSA, respectively (11). In comparison, the detection rates in our study region were relatively high, highlighting the importance and urgency of continuous monitoring of ocular drug-resistant bacteria.

Concurrently, the resistance of *Staphylococcus* to antibacterial agents was observed to increase with age, with a notably low resistance rate of 0% to levofloxacin among

patients under 24 years of age. This phenomenon can be attributed to the consideration that microorganisms, such as *Staphylococcus*, can asymptomatically colonize in patients, and systemic or local use of antibacterial agents for any infection type can increase the risk of colonization by resistant microorganisms at a low level, thereby predisposing patients to resistant microbial infections (5). Notably, quinolone antibacterial agents are prohibited for use in children, which has somewhat delayed the emergence of resistance to these agents in the pediatric population. Furthermore, our findings demonstrated a high level of consistency in drug sensitivity results among different generations of quinolone drugs. Therefore, we strongly recommend the rational use of antibacterial agents, particularly novel agents, to delay the emergence of antibacterial resistance.

Given the high resistance observed in gram-positive bacteria and fungi, we strongly recommend that clinicians routinely submit corneal scrapings from patients with MK for microbiological testing. Profiling pathogen resistance should guide immediate adjustments to medication based on susceptibility test results. Laboratories with appropriate capabilities are advised to perform *in vitro* susceptibility testing for filamentous fungi. The study limited to Southern Zhejiang reveals MK pathogen trends over 8 years, enhancing regional epidemiology. Given the widespread ocular pathogens, broadening isolate surveillance and establishing a nationwide resistance monitoring network are vital. This could help track regional or temporal variations and resistance profiles, provide information on targeted prevention, refine treatments, and safeguard patient vision.

## Conclusion

Data from our monitoring studies indicated trauma in men as a major risk factor for MK in the study region, with fungi, particularly filamentous fungi, being the primary pathogenic microorganisms. Mixed infections further increase the complexity of clinical treatment. Over the past 8 years, the overall level of resistance to antibacterial agents among microbial isolates from MK has remained stable. However, certain strains, such as MRS and MRSP, have exhibited a relatively high isolation rate. These results underscore the imperative of continuous observation and monitoring of resistance in these strains. Additionally, corneal *C. albicans* isolates exhibited a high level of resistance to voriconazole, whereas *A. flavus* demonstrated potential resistance to itraconazole and amphotericin B. Overall, our findings are crucial for guiding clinical treatment decisions and formulating strategies for monitoring and managing drug resistance. Given the regional scope of the study, a nationwide ocular bacterial resistance monitoring system is vital to transcending limitations, capturing broader data, and informing effective prevention and treatment strategies across the country.

## ACKNOWLEDGMENTS

We express our deepest gratitude to the cornea specialists at the Eye Hospital of Wenzhou Medical University for their invaluable support in providing the necessary data on keratitis pathogens. This contribution was not only critical to the successful completion of this study but also essential for gaining a deeper understanding of the evolution of resistance rates in pathogenic bacteria. This work was supported by the Science and Technology Plan Project of Wenzhou Municipality (nos. 2022ZY0042 and Y2023789). The authors declare that they have no known competing financial interests or personal relationships that could have influenced the work reported in this paper.

The original draft was written by Y.X. The methodology and formal analysis were developed and performed by Y.X., B.J.S., Y.Y.S., H.J.X., Y.F.G., L.P.M., Y.L., and Q.S.L. The manuscript was reviewed and edited by M.Q.Z.

## AUTHOR AFFILIATIONS

[1]National Clinical Research Center for Ocular Diseases, Eye Hospital, Wenzhou Medical University, Wenzhou, Zhejiang, China

[2]Wenzhou Key Laboratory of Sanitary Microbiology, Key Laboratory of Laboratory Medicine, School of Laboratory Medicine and Life Sciences, Ministry of Education, Wenzhou Medical University, Wenzhou, Zhejiang, China

## AUTHOR ORCIDs

Meiqin Zheng  http://orcid.org/0000-0003-3253-4480

## FUNDING

| Funder | Grant(s) | Author(s) |
| --- | --- | --- |
| Science and Technology Plan Project of Wenzhou | No. ZY2022018 | Meiqin Zheng |
| Science and Technology Plan Project of Wenzhou | No. Y2023789 | Yi Xu |

## AUTHOR CONTRIBUTIONS

Yi Xu, Conceptualization, Data curation, Formal analysis, Funding acquisition, Writing – original draft, Writing – review and editing | Bianjin Sun, Software | Yangyang Shen, Data curation, Formal analysis | Huijing Xu, Data curation, Formal analysis | Yunfeng Gu, Project administration | Liping Mao, Resources | Ying Liang, Validation | Qingsong Lu, Validation | Meiqin Zheng, Funding acquisition, Writing – review and editing

## DATA AVAILABILITY

The data sets used and/or analyzed during the current study are available from the corresponding author upon reasonable request.

## ADDITIONAL FILES

The following material is available online.

Open Peer Review

**PEER REVIEW HISTORY (review-history.pdf).** An accounting of the reviewer comments and feedback.

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
