## [Reviewer comments · Microbiology Spectrum]

Microbiology Spectrum

Etiological characteristics of 3691 cases of microbial keratitis: An 8-year longitudinal study

Yi Xu, Bianjin Sun, Yangyang Shen, Huijing Xu, Yunfeng Gu, Liping Mao, Ying Liang, Qingsong Lu, and Meiqin Zheng

Corresponding Author(s): Meiqin Zheng, Wenzhou Medical University Eye Hospital

Review Timeline:

Submission Date:	October 23, 2024
Editorial Decision:	December 9, 2024
Revision Received:	January 6, 2025
Accepted:	March 3, 2025

Editor: Yi-Chin Fan

Reviewer(s): Disclosure of reviewer identity is with reference to reviewer comments included in decision letter(s). The following individuals involved in review of your submission have agreed to reveal their identity: Innocent Afeke (Reviewer #1); suleiman fawaz al obeid (Reviewer #2)

Transaction Report:

DOI: <https://doi.org/10.1128/spectrum.02630-24>

Re: Spectrum02630-24 (An 8-year longitudinal study of the etiological characteristics of 3691 cases of microbial keratitis)

Dear Dr. Meiqin Zheng:

Thank you for the privilege of reviewing your work. Below you will find my comments, instructions from the Spectrum editorial office, and the reviewer comments.

Revision Guidelines

Sincerely,
Yi-Chin Fan
Editor
Microbiology Spectrum

Reviewer #1 (Comments for the Author):

This study addresses a critical area in ophthalmic microbiology by analyzing long-term data to identify trends in the etiological agents and their clinical implications. I have the following concerns.

(a) Abstract

1. The abbreviation MK first used in the abstract should be written in full.

2. The abstract does not specify the study's geographical or demographic context. Adding this information would enhance its relevance and contextual understanding.
3. The authors should briefly state whether this rate of 38.72% is high, typical, or low compared to other regions or similar studies.
4. The authors should add a concluding sentence summarizing the study's implications for clinical practice or future research.

(b) Importance of the study

The authors should specifically state the gap in knowledge in the sciences that their study has addressed. What is known, what is unknown, and what has your study contributed to the study area?

(c) Background

1. The authors should include regional or local data (if available) to better contextualize MK's significance in the study area.
2. The necessity of antimicrobial susceptibility testing for treatment is standard knowledge in the field. The authors should focus on unique challenges or insights related to MK management in the study region.
3. Some aspects of the study objectives are unclear (Lines 78-83). For example, how does the study aim to address the gaps in resistance monitoring systems? I suggest the authors state the study's specific aims, such as identifying resistance trends, evaluating treatment efficacy, or proposing new clinical guidelines.

(d) Methods

The Methods section is well-structured. However, I have the following concerns.

1. Line 89-the authors stated that they include patients with confirmed positive bacterial and fungal cultures in their study. *Acanthamoeba* is reported in the abstract and another part of the manuscript. Is it also a bacterium or fungus?
2. Line 102: What criteria did the authors use to classify some cultures' suspected contaminated strains?
3. I suggest the authors list the antibiotics and antifungals used for susceptibility testing to provide a complete picture of the examined resistance profiles.

(e) Results

A summary of the most critical findings at the end of the Results section could help readers synthesize the key points.

(f) Discussion

Authors should discuss how the identified resistance patterns and pathogen prevalence could influence local treatment guidelines, including the selection of empirical therapies and the importance of susceptibility testing.

Reviewer #2 (Comments for the Author):

English terms and expressions need to be improved throughout the whole manuscript.

I think it is better to be concentrated the work on the 1382 cases, then create a table concerning those harbored mixed agent of infections 55 cases to reflect the protocol of the treatment EX: bacterial infection (gram positive and gram negative and their susceptibility) or (bacterial infection with fungal infection). please revise the numbers presented in the table 1

**An 8-year longitudinal study of the etiological characteristics of 3691 cases of**
**microbial keratitis**

**Authors**

Yi Xu¹, Bianjin Sun², Yangyang Shen¹, Huijing Xu¹, Yunfeng Gu¹, Liping Mao¹, Ying
Liang¹, Qingsong Lu¹, Meiqin Zheng^{1#}

**Affiliations**

¹National Clinical Research Center for Ocular Diseases, Eye Hospital, Wenzhou
Medical University, Wenzhou 325027, China

²Wenzhou Key Laboratory of Sanitary Microbiology, Key Laboratory of Laboratory
Medicine, School of Laboratory Medicine and Life Sciences, Ministry of Education,
Wenzhou Medical University, Wenzhou 325035, Zhejiang, China

[#]Correspondence:

Meiqin Zheng

email: zmq@eye.ac.cn

TEL: 86-577-88068857

**Abstract**

This study encompassed 3,691 patients with MK and revealed a 38.72% positivity rate
in microbial cultures. Trauma and foreign body entry emerged as significant risk

factors. Mixed infections, accounting for 4.85%, posed challenges for diagnosis and
treatment. Filamentous fungi, predominantly *Fusarium* spp., dominated the microbial
landscape. Prominent bacterial pathogens included *Pseudomonas aeruginosa*,
*Streptococcus pneumoniae*, *Staphylococcus epidermidis*, and *Staphylococcus aureus*.
*Acanthamoeba spinosa* was an important pathogen affecting the cornea in this region.
The observed resistance patterns emphasized the urgency for alternative therapeutic
strategies targeting resistant gram-positive (e.g., erythromycin, penicillin, oxacillin)
and gram-negative (e.g., trimethoprim-sulfamethoxazole, tetracycline) bacteria as well
as refractory fungi such as *Candida albicans* (voriconazole-resistant) and *Aspergillus*
*flavus* (resistant to itraconazole and amphotericin B). Resistance to ceftazidime,
meropenem, and erythromycin exhibited a slight upward trend, diverging from the
overall bacterial resistance trend observed in China. High isolation rates of
methicillin-resistant *S. aureus* and macrolide-resistant *S. pneumoniae* underscored the
need for enhanced infection control measures and targeted interventions against these
resistant pathogens. The consistent susceptibility outcomes across multiple generations
of quinolone antibiotics highlight the importance of prudent and rational use of
antimicrobial agents.

**Importance:**

This study aimed to analyze the distribution of pathogenic microorganisms and their
antibiotic-resistance profiles in microbial keratitis (MK) cases in the southern region of
Zhejiang Province, China, between 2016 and 2023. Culture results were systematically

collected from patients diagnosed with MK over the specified period, and a rigorous
retrospective analysis was conducted, encompassing pathogen identification and
comprehensive drug sensitivity testing for all positive specimens. Overall, in southern
Zhejiang, filamentous fungi, notably *Fusarium* spp., emerged as the primary MK
pathogens. The results of drug-resistance monitoring indicate that continuous
monitoring is essential. Timely intervention after trauma or foreign body entry is
essential to reduce the risk of infection; clinicians must increase culture submissions
and tailor antimicrobial therapy based on culture results to ensure safe, effective, and
sustainable treatments.

**Keywords:** Keratitis; Eye Trauma; Filamentous Fungi; Multidrug Resistance

**1. Background**

The terms 'microbial keratitis', 'infective keratitis', and 'suppurative keratitis' refer to
suppurative infections of the cornea (1). Microbial keratitis (MK) is an ocular
emergency primarily caused by bacteria, fungi, viruses, or protozoa and is
characterized by a high incidence rate and poor visual prognosis. This condition is
prevalent worldwide, particularly in South, Southeast, and East Asia, with an estimated
annual incidence exceeding 2 million cases. MK is the fifth leading cause of blindness
globally, accounting for more than 5% of all blindness cases (2). The latency period of
MK ranges from 2 to 30 days, with the disease progressing rapidly after onset. Clinical

manifestations include corneal ulcers, epithelial defects, and interstitial infiltration.
Without prompt and effective treatment, MK can progress swiftly, leading to corneal
perforation, eye atrophy, blindness, and other adverse outcomes that severely affect the
daily lives and work of patients. Therefore, aggressive treatment and prompt control of
MK are crucial for improving patient prognosis (3). Bacteria and fungi represent the
primary pathogens of MK (2); culturing can facilitate the identification of specific
causative agents. Antimicrobial susceptibility testing provides a basis for antibiotic
treatment, and statistical analysis of these culture results can help reveal resistant
strains, assess resistance trends, and provide a scientific basis for the rational use of
antibiotics in clinical practice. The human ocular system exhibits a complex structure,
with significant differences in the spectra of pathogens isolated from distinct
physiological structures (4). However, existing domestic and international resistance
monitoring systems predominantly offer broad analyses of ocular pathogens, which are
inadequate for addressing the specific demands of clinical practice. Therefore, this
study was undertaken to analyze the distribution and resistance of pathogens
responsible for MK in southern Zhejiang over the past 8 years to provide valuable
insights for clinical treatment strategies.

**2. Methods**

**2.1. Sample Source**

[revised manuscript text omitted]

570 R: rate of drug resistance; I: rate of drug intermediates; S: rate of drug sensitivity.

571

572

574 Table 1. Predisposing factors of patients with microbial keratitis.

Predisposing factors	Occupancy	Gram-positive	Gram-negative	Filamentous	Yeast	Acanthamoeba
	n=1382; n (%)	bacteria n=340; n (%)	bacteria n=259; n (%)	fungi n=763; n (%)	n=94; n (%)	spinososa n=13; n (%)
Traumatic	721 (52.17)	130 (38.24)	114 (44.02)	470 (61.60)	27 (28.72)	5 (38.46)
Plant-related	307 (22.21)	38 (11.18)	32 (12.36)	235 (30.80)	9 (9.57)	1 (7.69)
Metal-related	92 (6.66)	17 (5.00)	25 (9.65)	48 (6.29)	5 (5.32)	0 (0)
Hand-related	27 (1.95)	7 (2.06)	8 (3.09)	12 (1.57)	1 (1.06)	1 (7.69)
Others ^a	295 (21.35)	68 (20.00)	49 (18.92)	175 (22.94)	12 (12.77)	3 (23.08)
Ophthalmic surgeries	111 (8.03)	36 (10.59)	20 (7.72)	25 (3.28)	37 (39.36)	0 (0)
Post-keratoplasty	68 (4.92)	18 (5.29)	12 (4.63)	15 (1.97)	28 (29.79)	0 (0)
Others ^b	43 (3.11)	18 (5.29)	8 (3.09)	10 (1.31)	9 (9.57)	0 (0)
Ocular diseases ^c	61 (4.41)	24 (7.06)	10 (3.86)	23 (3.01)	8 (8.51)	0 (0)
Contact lens-related	17 (1.23)	8 (3.37)	7 (2.70)	2 (0.26)	0 (0)	2 (14.94)

Unknown	472 (34.15)	142 (41.76)	108 (41.70)	243 (31.85)	22 (23.40)	6 (44.81)
---------	-------------	-------------	-------------	-------------	---------------	-----------

575 a: Sand, dust, insects, chemicals, etc.

b: Cataract surgery, penetrating keratoplasty, pterygium excision, etc.

c: Exposure keratitis, corneal dystrophy, etc.

**Table 2.** Minimum inhibitory concentrations and resistance profiles for corneal

bacterial isolates from 2016 to 2023.

Antimicrobials	Gram-positive bacteria (n=340)	S. epidermidi s (n=63)	S. aureus (n=43)	S. pneumonia (n=64)	Gram-negati ve bacteria (n=259)	S. aeruginos a (n=73)
Penicillin G,R* (MIC ₅₀)	58.5 (1)	98.2 (8)	91.7 (8)	4.16 (1)	N/A	N/A
Piperacillin- Tazobactam, R (MIC ₅₀)	N/A	N/A	N/A	N/A	14.0 (8/4)	8.6 (8/4)
Oxacillin, R (MIC ₅₀)	45.5 (1)	57.1 (1)	20.9 (0.5)	N/A	N/A	N/A
Ceftriaxone, R (MIC ₅₀)	5.8 (0.5)	N/A	N/A	0.0 (0.5)	20.0 (0.25)	N/A
Ceftazidime, R (MIC ₅₀)	N/A	N/A	N/A	N/A	11.8 (2)	4.4 (2)
Meropenem, R (MIC ₅₀)	N/A	N/A	N/A	N/A	4.2 (0.25)	0.0 (0.5)
Erythromycin, R (MIC ₅₀)	71.7 (16)	67.4 (16)	62.1 (8)	90.5 (32)	N/A	N/A
Cotrimoxazole, R (MIC ₅₀)	23.4 (0.25/4.75)	N/A	N/A	N/A	44.4 (2/38)	N/A

Ofloxacin, R (MIC ₅₀)	53.3 (8)	69.5 (8)	38.5 (1)	11.1 (1)	14.6 (1)	11.5 (1)
Levofloxacin, R (MIC ₅₀)	30.9 (1)	54.0 (4)	30.6 (0.5)	0.0 (1)	10.5 (0.5)	4.8 (0.5)
Gatifloxacin, R (MIC ₅₀)	24.0 (0.5)	38.5 (1)	N/A	0.0 (0.25)	12.0 (0.5)	N/A
Clindamycin, R (MIC ₅₀)	53.7 (2)	N/A	N/A	N/A	N/A	N/A
Gentamicin, R (MIC ₅₀)	50.0 (4)	37.0 (8)	34.4 (1)	N/A	17.4 (1)	6.8 (2)
Amikacin, R (MIC ₅₀)	8.3 (4)	N/A	N/A	N/A	11.7 (4)	1.8 (4)
Tobramycin, R (MIC ₅₀)	46.8 (16)	47.6 (8)	42.9 (8)	N/A	21.1 (2)	6.9 (1)
Chloramphenicol, R (MIC ₅₀)	11.4 (4)	24.6 (8)	9.1 (8)	6.7 (2)	29.8 (8)	0.0 (32)
Vancomycin, R (MIC ₅₀)	2.3 (1)	0.0 (2)	0.0 (1)	0.0 (2)	N/A	N/A
Linezolid, R (MIC ₅₀)	0.0 (1)	0.0 (1)	0.0 (1)	N/A	N/A	N/A
Tetracycline, R (MIC ₅₀)	45.0 (4)	31.3 (4)	22.7 (0.5)	70.5 (16)	50.0 (1)	N/A

582 R*: rate of drug resistance (%); N/A = not applicable; MIC₅₀: the minimum inhibitory
concentration (µg/mL) that inhibited 50% of the tested microorganisms.

**Table 3.** Minimum inhibitory concentrations and resistance profiles for corneal fungal isolates from 2016 to 2023.

		C. Albicans	C. parapsilosis	Other yeast	P -value	Fusarium spp.	A. flavus	A. fumigatus	Other filamentous fungi	p -value
		n=13	n=48	n=33		n=314	n=49	n=35	n=365	
Antimicrobials	R* (%)	46.2 (6/13)	0 (0/48)	5.9	0.043	N/A	N/A	2.9 (1/35)	N/A	< 0.001
	Voriconazole MIC ₅₀ (µg/mL)	0.19	0.016	0.032		1	0.125	0.125	0.19	
	MIC ₉₀ (µg/mL)	64	0.064	0.75		8	0.5	0.38	1.5	
	Itraconazole R (%)	N/A	N/A	N/A	0.043	N/A	16.3	32.4	N/A	< 0.001

								(11/34)		
	MIC ₅₀ (µg/mL)	0.19	0.19	0.25		64	0.5	0.75	0.75	
	MIC ₉₀ (µg/mL)	32	0.25	1.5		64	4	4	64	
	R (%)	N/A	N/A	N/A	0.049	N/A	N/A	42.9 (15/35)	N/A	< 0.001
Amphotericin	MIC ₅₀ (µg/mL)	0.25	0.25	0.75		1	1.5	1	0.5	
	MIC ₉₀ (µg/mL)	0.75	1	64		64	32	8	64	

587 R*: rate of drug resistance; N/A = not applicable; MIC₅₀ and MIC₉₀ are defined as the minimum inhibitory concentrations that inhibited 50% and
588 90% of the tested microorganisms, respectively; significant *P*-values are indicated in bold.

- Ofloxacin
- Levofloxacin
- Gatifloxacin
- Tobramycin
- Ceftriaxime
- Oxacillin
- Meropenem
- Amikacin
- Gentamicin
- Tobramycin
- Ceftriaxime
- Vancomycin
- Erythromycin
- Chloramphenicol
- Voriconazole

- Ofloxacin
- Levofloxacin
- Gentamicin
- Tobramycin
- Vancomycin
- Penicillin G
- Cefazidime
- Ceftriaxone
- Oxacillin
- Meropenem
- Erythromycin
- Chloramphenicol

Staphylococcus spp.

a

Levofloxacin efficacy consistency gatifloxacin

● Gatifloxacin R ● Levofloxacin S ● Levofloxacin I
● Levofloxacin R ● Gatifloxacin I ● Gatifloxacin S

b

Levofloxacin efficacy consistency ofloxacin

● Ofloxacin R ● Levofloxacin S ● Levofloxacin I
● Levofloxacin R ● Ofloxacin I ● Ofloxacin S

Dear Dr. Fan

We sincerely appreciate the meticulous review and constructive feedback provided by the reviewers. Their valuable insights have been instrumental in refining our manuscript.

We have incorporated the reviewers' suggestions to further enhance the quality of our work.

Reviewers, Comments to the Authors:

Reviewer#1

This study addresses a critical area in ophthalmic microbiology by analyzing long-term data to identify trends in the etiological agents and their clinical implications.

Author response: Thank you for your kind words and thoughtful feedback. We are delighted that you recognize the importance of our work in addressing this critical area in ophthalmic microbiology.

(a) Abstract

1. The abbreviation MK first used in the abstract should be written in full.

Author response: Thank you for your valuable comment. We have added the full term for microbial keratitis (MK) before its first use as an abbreviation.

2. The abstract does not specify the study's geographical or demographic context. Adding this information would enhance its relevance and contextual understanding.

Author response: Thank you for your insightful comment. The description has been revised to specify that the patient data originates from the southern Zhejiang region. Following meticulous deduplication, 1,382 unique cases were identified for analysis. The results revealed a higher proportion of male patients, with a sex ratio of 2.43:1 (979 men vs. 403 women). The ages of the patients ranged from 2 to 93 years, with a mean age of 58.6 ± 14.7 years.

3. The authors should briefly state whether this rate of 38.72% is high, typical, or low compared to other regions or similar studies.

Author response: Thank you for pointing this out. A review of relevant domestic literature revealed a lack of cultivation data. Compared to other regions globally, our positive cultivation rate is relatively low (1). This observation is highlighted in the abstract, and a detailed cause analysis is provided in the Discussion section.

4. The authors should add a concluding sentence summarizing the study's implications for clinical practice or future research.

Author response: Thank you for bringing this to our attention. A concluding sentence summarizing the study's clinical implications has been added to the end of the abstract.

(b) Importance of the study

The authors should specifically state the gap in knowledge in the sciences that their study has addressed. What is known, what is unknown, and what has your study contributed to the study area?

Author response: We appreciate your attention to this matter. We have revised the entire section on the importance of the study to more effectively highlight its significance and value. The updated version emphasizes the critical need for this research in addressing a substantial gap in understanding antibiotic resistance trends among ocular isolates, particularly those associated with corneal infections, in our region.

(c) Background

1. The authors should include regional or local data (if available) to better contextualize MK's significance in the study area.

Author response: Thank you for your perceptive comment. Currently, our region lacks comprehensive, long-term, large-scale data on the infection patterns and drug resistance surveillance of corneal isolates. The etiological spectrum of MK exhibits considerable regional variability (1). This information has been added to the Introduction section.

2. The necessity of antimicrobial susceptibility testing for treatment is standard knowledge in the field. The authors should focus on unique challenges or insights related to MK management in the study region.

Author response: Thank you for your thoughtful comment. We have removed certain common-sense statements, such as those explaining how culturing identifies specific causative agents, how antimicrobial susceptibility underpins antibiotic treatment, and how statistical analysis of culture results aids in uncovering resistant strains, assessing resistance trends, and providing a scientific basis for the judicious use of antibiotics in clinical settings. This adjustment refocuses the content more directly on the management of MK.

3. Some aspects of the study objectives are unclear (Lines 78-83). For example, how does the study aim to address the gaps in resistance monitoring systems? I suggest the authors state the study's specific aims, such as identifying resistance trends, evaluating treatment efficacy, or proposing new clinical guidelines.

Author response: Thank you for your important comment. We have refined the expressions in lines 71 to 81 to clearly delineate the research aims, specifically focusing on the surveillance of antibiotic resistance trends and the development of therapeutic guidance.

(d) Methods

The Methods section is well-structured. However, I have the following concerns.

1. Line 89-the authors stated that they include patients with confirmed positive bacterial and fungal cultures in their study. Acanthamoeba is reported in the abstract and another part of the manuscript. Is it also a bacterium or fungus?

Author response: Thank you for your meaningful comment. We have corrected the erroneous statement by explicitly clarifying that *Acanthamoeba* is a parasite.

2. Line 102: What criteria did the authors use to classify some cultures' suspected contaminated strains?

Author response: Thank you for your valuable question. We have included criteria in the article for excluding contamination. Cultures were classified as suspected contaminated strains based on the following criteria: colonies failing to grow along the inoculation streak, and isolated strains not correlating with clinical infection symptoms or results from auxiliary tests such as confocal microscopy and smear examination.

3. I suggest the authors list the antibiotics and antifungals used for susceptibility testing to provide a complete picture of the examined resistance profiles.

Author response: Thank you for your valuable suggestion. We have provided supplementary explanations, stating that, based on the CLSI M100 and M45 guidelines as well as standard ophthalmic medications, our department has established over 20 antibiotic susceptibility testing combinations tailored to the species of the isolated bacteria.

(e) Results

A summary of the most critical findings at the end of the Results section could help readers synthesize the key points.

Author response: Thank you for your insightful suggestion. We have added a summary at the end of the Results section: "Our study identified trauma and the introduction of foreign bodies as key predisposing factors for MK in our region, with *Fusarium* being the primary causative agent. Mixed infections pose significant challenges to the clinical management of MK. Notably, gram-positive bacteria, particularly *Staphylococcus spp.*, exhibit high resistance to quinolones, aminoglycosides, and β -lactams, with resistance patterns correlating with patient age. Continuous monitoring has revealed a slight increase in resistance to cephalosporins, including ceftazidime, meropenem, and erythromycin. The potential for fungal resistance in the treatment of fungal keratitis warrants clinical attention."

(f) Discussion

Authors should discuss how the identified resistance patterns and pathogen prevalence could

influence local treatment guidelines, including the selection of empirical therapies and the importance of susceptibility testing.

Author response: Your meaningful comment is duly noted. Based on the observed resistance patterns and the prevalence of pathogens, we have included the following professional treatment recommendations: “Given the high resistance observed in gram-positive bacteria and fungi, we strongly recommend that clinicians routinely submit corneal scrapings from patients with MK for microbiological testing. Profiling pathogen resistance should guide immediate adjustments to medication based on susceptibility test results. Laboratories with appropriate capabilities are advised to perform *in vitro* susceptibility testing for filamentous fungi.”

Reviewer #2:

1.English terms and expressions need to be improved throughout the whole manuscript.

Author response: Thank you for your perceptive suggestion, and we apologize for this issue. Some of the longer sentences have been shortened and reorganized. To enhance conciseness, we have deleted certain content. We also used the services of a professional editor to improve the language of the manuscript.

2.I think it is better to be concentrated the work on the 1382 cases, then create a table concerning those harbored mixed agent of infections 55 cases to reflect the protocol of the treatment EX: bacterial infection (gram positive and gram negative and their susceptibility) or (bacterial infection with fungal infection).

Author response: Your insightful comment is recognized. We have conducted a fresh statistical analysis of the mixed infection data and created Table 2. These cases have been categorized into the following groups: bacterial-only mixed infections, bacterial-fungal mixed infections, and fungal-only mixed infections. Among the 67 mixed infections, 36 were identified as bacterial, involving species such as *Streptococcus* spp., *Staphylococcus* spp., *Corynebacterium* spp., *Haemophilus* spp., and *Chryseobacterium* spp. An additional 26 cases were classified as bacterial-fungal mixed infections, predominantly involving *Staphylococcus* spp., *Fusarium* spp., and *Candida* spp. Five cases were categorized as fungal-only infections, primarily involving *Candida* spp. in conjunction with filamentous fungi. The collected data have been analyzed in the Discussion section to effectively inform the treatment protocol.

Table 2.Composition of 67 cases of mixed infection.

Category	Bacterial-only mixed infection, n (%) (N=36)	Bacterial-fungal mixed infection, n (%) (N=26)	Fungal-only mixed infection, n (%) (N=5)
----------	--	--	--

Main isolate	Streptococcus spp. 15 (41.67)	Staphylococcus spp. 12 (46.15)	Candida spp. 5 (100.00)
	Staphylococcus spp. 8 (22.22)	Fusarium spp. 9 (34.62)	Filamentous fungi 4 (80.00)
	Corynebacterium spp. 7 (26.92)	Candida spp. 5 (19.23)	
	Haemophilus spp. 6 (23.08)		
	Chryseobacterium spp. 5 (19.23)		

n: individuals infected by a specific microbe; N: total individuals with mixed infections.

3. please revise the numbers presented in the table 1

Author response: Your insightful comment is appreciated. We have revised the numerical values in Table 1 and clarified any ambiguous terminology to ensure that the tabular data accurately convey our observations.

We sincerely thank the reviewers for their time and dedication in assessing our manuscript. We welcome any additional feedback or suggestions that could help us further enhance it.

References

1. Ung L, Bispo PJM, Shanbhag SS, Gilmore MS, Chodosh J. 2019. The persistent dilemma of microbial keratitis: global burden, diagnosis, and antimicrobial resistance. *Surv Ophthalmol* 64:255–271.

Re: Spectrum02630-24R1 (Etiological characteristics of 3691 cases of microbial keratitis: An 8-year longitudinal study)

Dear Dr. Meiqin Zheng:

Your manuscript has been accepted, and I am forwarding it to the ASM production staff for publication. Your paper will first be checked to make sure all elements meet the technical requirements. ASM staff will contact you if anything needs to be revised before copyediting and production can begin. Otherwise, you will be notified when your proofs are ready to be viewed.

Sincerely,
Yi-Chin Fan
Editor
Microbiology Spectrum

Reviewer #1 (Comments for the Author):

The authors have adequately addressed my concerns. Gram is a name of a person so should start with the capital G.

Reviewer #2 (Comments for the Author):

Dear authors,
thank you for reviewing our comments and answering in a precise scientific manner. we do not have any other observation.

thank you

Etiological characteristics of 3691 cases of microbial keratitis: An 8-year longitudinal study

Running title: An 8-year study on microbial keratitis etiology

Authors

Yi Xu¹, Bianjin Sun², Yangyang Shen¹, Huijing Xu¹, Yunfeng Gu¹, Liping Mao¹, Ying Liang¹, Qingsong Lu¹, Meiqin Zheng^{1#}

Affiliations

¹National Clinical Research Center for Ocular Diseases, Eye Hospital, Wenzhou Medical University, Wenzhou 325027, China

²Wenzhou Key Laboratory of Sanitary Microbiology, Key Laboratory of Laboratory Medicine, School of Laboratory Medicine and Life Sciences, Ministry of Education, Wenzhou Medical University, Wenzhou 325035, Zhejiang, China

[#]Correspondence: Meiqin Zheng

email: zmq@eye.ac.cn

TEL: 86-577-88068857

Word count for the abstract: 248

Word count for the text: 4,312

[revised manuscript text omitted]

		C. Albicans	C. parapsilosis	Other yeast	P-value	Fusarium spp.	A. flavus	A. fumigatus	Other filamentous fungi	P-value
		n=13	n=48	n=33		n=314	n=49	n=35	n=365	
Antimicrobials	R* (%)	46.2 (6/13)	0 (0/48)	5.9	0.043	N/A	N/A	2.9 (1/35)	N/A	< 0.001
	Voriconazole MIC ₅₀ (µg/mL)	0.19	0.016	0.032		1	0.125	0.125	0.19	
	Voriconazole MIC ₉₀ (µg/mL)	64	0.064	0.75		8	0.5	0.38	1.5	
	Itraconazole R (%)	N/A	N/A	N/A	0.043	N/A	16.3	32.4	N/A	< 0.001

								(11/34)		
	MIC ₅₀ (µg/mL)	0.19	0.19	0.25		64	0.5	0.75	0.75	
	MIC ₉₀ (µg/mL)	32	0.25	1.5		64	4	4	64	
	R (%)	N/A	N/A	N/A	0.049	N/A	N/A	42.9 (15/35)	N/A	< 0.001
Amphotericin	MIC ₅₀ (µg/mL)	0.25	0.25	0.75		1	1.5	1	0.5	
	MIC ₉₀ (µg/mL)	0.75	1	64		64	32	8	64	

R*: rate of drug resistance; N/A = not applicable; MIC₅₀ and MIC₉₀ are defined as the minimum inhibitory concentrations that inhibited 50% and 90% of the tested microorganisms, respectively; significant *P*-values are indicated in bold.

Etiological characteristics of 3691 cases of microbial keratitis: An 8-year longitudinal study

Running title: An 8-year study on microbial keratitis etiology

Authors

Yi Xu¹, Bianjin Sun², Yangyang Shen¹, Huijing Xu¹, Yunfeng Gu¹, Liping Mao¹, Ying Liang¹, Qingsong Lu¹, Meiqin Zheng^{1#}

Affiliations

¹National Clinical Research Center for Ocular Diseases, Eye Hospital, Wenzhou Medical University, Wenzhou 325027, China

²Wenzhou Key Laboratory of Sanitary Microbiology, Key Laboratory of Laboratory Medicine, School of Laboratory Medicine and Life Sciences, Ministry of Education, Wenzhou Medical University, Wenzhou 325035, Zhejiang, China

[#]Correspondence: Meiqin Zheng

email: zmq@eye.ac.cn

TEL: 86-577-88068857

Word count for the abstract: 248

Word count for the text: 4,312

[revised manuscript text omitted]

		C. Albicans	C. parapsilosis	Other yeast	P-value	Fusarium spp.	A. flavus	A. fumigatus	Other filamentous fungi	P-value
		n=13	n=48	n=33		n=314	n=49	n=35	n=365	
Antimicrobials	R* (%)	46.2 (6/13)	0 (0/48)	5.9	0.043	N/A	N/A	2.9 (1/35)	N/A	< 0.001
	Voriconazole MIC ₅₀ (µg/mL)	0.19	0.016	0.032		1	0.125	0.125	0.19	
	Voriconazole MIC ₉₀ (µg/mL)	64	0.064	0.75		8	0.5	0.38	1.5	
	Itraconazole R (%)	N/A	N/A	N/A	0.043	N/A	16.3	32.4	N/A	< 0.001

								(11/34)		
	MIC ₅₀ (µg/mL)	0.19	0.19	0.25		64	0.5	0.75	0.75	
	MIC ₉₀ (µg/mL)	32	0.25	1.5		64	4	4	64	
	R (%)	N/A	N/A	N/A	0.049	N/A	N/A	42.9 (15/35)	N/A	< 0.001
Amphotericin	MIC ₅₀ (µg/mL)	0.25	0.25	0.75		1	1.5	1	0.5	
	MIC ₉₀ (µg/mL)	0.75	1	64		64	32	8	64	

R*: rate of drug resistance; N/A = not applicable; MIC₅₀ and MIC₉₀ are defined as the minimum inhibitory concentrations that inhibited 50% and 90% of the tested microorganisms, respectively; significant *P*-values are indicated in bold.